# PRCA: Fitting Black-Box Large Language Models for Retrieval Question Answering via Pluggable Reward-Driven Contextual Adapter

**Haoyan Yang[1,2†], Zhitao Li[1], Yong Zhang[1], Jianzong Wang[1*],**
**Ning Cheng[1], Ming Li[1,3], Jing Xiao[1]**

[1]Ping An Technology (Shenzhen) Co., Ltd., China
[2]New York University  [3]University of Maryland
jzwang@188.com

## Abstract

The Retrieval Question Answering (ReQA) task employs the retrieval-augmented framework, composed of a retriever and generator. The generator formulates the answer based on the documents retrieved by the retriever. Incorporating Large Language Models (LLMs) as generators is beneficial due to their advanced QA capabilities, but they are typically too large to be fine-tuned with budget constraints while some of them are only accessible via APIs. To tackle this issue and further improve ReQA performance, we propose a trainable **P**luggable **R**eward-Driven **C**ontextual **A**dapter (PRCA), keeping the generator as a black box. Positioned between the retriever and generator in a Pluggable manner, PRCA refines the retrieved information by operating in a token-autoregressive strategy via maximizing rewards of the reinforcement learning phase. Our experiments validate PRCA's effectiveness in enhancing ReQA performance on three datasets by up to 20% improvement to fit black-box LLMs into existing frameworks, demonstrating its considerable potential in the LLMs era.

## 1 Introduction

Retrieval Question Answering (ReQA) tasks involve generating appropriate answers to given questions, utilizing relevant contextual documents. To achieve this, retrieval augmentation is employed (Chen et al., 2017; Pan et al., 2019; Izacard and Grave, 2021), and comprised of two key components: a retriever and a generator. The retriever's role is to retrieve relevant documents from a large corpus in response to the question, while the generator uses this contextual information to formulate accurate answers. Such systems alleviate the problem of hallucinations (Shuster et al., 2021), thereby enhancing the overall accuracy of the output.

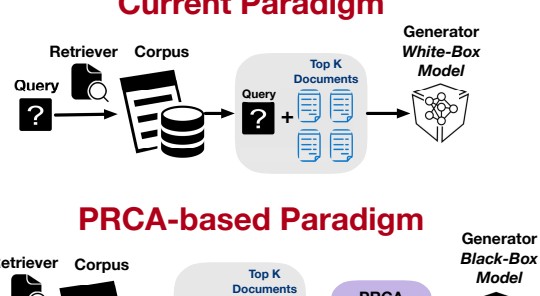

Figure 1: A comparison between two paradigms for information retrieval and generation. The upper section showcases the traditional method where a query is processed by a retriever that scans a corpus to fetch the Top-K documents and then fed to a white-box generator. The lower section introduces our proposed PRCA method, which processes extracted Top-K documents from the retriever before feeding them to black-box generator to achieve better performance for in-domain tasks.

Recent advances in Large Language Models (LLMs) such as the generative pre-trained transformer (GPT) series (Brown et al., 2020; Ouyang et al., 2022; OpenAI, 2023) have demonstrated remarkable potential, notably in their zero-shot and few-shot abilities within the realm of QA tasks. Owing to these capabilities, LLMs are excellent choices as generators within the retrieval-augmented framework. However, due to the vast parameters of LLMs, fine-tuning them becomes exceedingly difficult within a limited computation budget. Furthermore, certain LLMs such as GPT-4 (OpenAI, 2023) are closed-source, making it impossible to fine-tune them. To achieve optimal results on specific datasets, fine-tuning retrieval-augmented models becomes necessary (Guu et al., 2020; Lewis et al., 2020b; An et al., 2021). Previous attempts to integrate LLMs into the retrieval-augmented framework have met with partial suc-

---

[†] The work was done when the first author was doing internship at Ping An Technology (Shenzhen) Co., Ltd., China.
[*]Corresponding author: Jianzong Wang.

cess but also come with limitations. (Shi et al., 2023) utilized the logits from the final layer of the LLMs when calculating the loss function, which may not be available to certain powerful LLMs that served via APIs. (Ma et al., 2023) involved frequently invoking pricy LLMs and overlooked the impact of the input token length on the accuracy and effectiveness of the system.

To overcome these hurdles, we propose a trainable Pluggable Reward-driven Context Adapter (PRCA) that enables one to fine-tune the adapter instead of LLMs under the retrieval-augmented framework on specific datasets and achieve higher performance. Furthermore, PRCA distills the retrieved documents information guided by rewards from the generator through reinforcement learning. The distillation of retrieval information through PRCA reduces the length of text input to the generator and constructs a context of superior quality, which mitigates the hallucination issues during the answer generation. As shown in Figure 1, PRCA is placed between the retriever and the generator, forming a PRCA-based Paradigm where both the generator and the retriever remain frozen. In general, the introduction of the PRCA-based paradigm brings the following advantages:

**Black-box LLMs Integration** With the use of PRCA, LLMs can be treated as a black box integrated into the retrieval-augmented framework, eliminating the need for resource-intensive fine-tuning and restrictions on closed-nature models.

**Robustness** PRCA serves as a pluggable adapter that is compatible with various retrievers and generators because PRCA-based paradigm keeps both the generator and retriever frozen.

**Efficiency** The PRCA-based paradigm ensures the efficiency of the framework by reducing the text length inputted into the generator and can adapt to different retrieval corpus.

## 2  Related Work

### 2.1  The Potential of LLMs as Black-Box Models

LLMs have demonstrated remarkable capabilities in downstream QA tasks, even in scenarios with limited or no training data (Wei et al., 2022). This emergence capability enables them to efficiently tackle such tasks, making them potential candidates

for black-box models in inference. Furthermore, the non-open-source nature and large parameter size of these models further contribute to their inclination towards being perceived as black boxes.

On one hand, LLMs like GPT-4 (OpenAI, 2023) and PaLM (Scao et al., 2023) have showcased impressive performance in QA tasks. However, their closed source nature restricts access to these models, making API-based utilization the only feasible option, thereby categorizing them as black-box models.

On the other hand, training LLMs, exemplified by models like Bloom (Scao et al., 2022) and GLM-130B (Zeng et al., 2023), impose substantial computational demands. Specifically, training Bloom took 3.5 months using 384 NVIDIA A100 80GB GPUs. Similarly, GLM-130B requires a two-month training period on a cluster of 96 DGX-A100 GPU servers. These resource requirements make it extremely challenging for the majority of researchers to deploy these models. Moreover, LLMs exhibit rapid development speeds. For instance, from LLaMA (Touvron et al., 2023) to Alpaca (Taori et al., 2023) and now Vicuna (Peng et al., 2023), the iterations are completed within a month. It is evident that the speed of training models lags behind the pace of model iterations. Consequentially, tuning small-size adapters for any sequence-to-sequence LLMs on downstream tasks could be a simpler and more efficient approach.

### 2.2  Retrieval-Augmented Framework

Various retrieval augmented ideas have been progressively developed and applied to improve the performance in the ReQA task.

In the initial stage of research, independent statistical similarity-base retrievers like TF-IDF (Sparck Jones, 1972) and BM25 (Robertson and Zaragoza, 2009) were used as fundamental retrieval engines. They helped in extracting the most relevant documents from the corpus for QA tasks (Chen et al., 2017; Izacard and Grave, 2021).

The concept of vectorization was subsequently introduced, where both questions and documents were represented as vectors, and vector similarity became a critical parameter for retrieval. This paradigm shift was led by methods such as dense retrieval, as embodied by DPR (Karpukhin et al., 2020). Models based on contrastive learning like SimCSE (Gao et al., 2021) and Contriver (Izacard et al., 2022a), along with sentence-level se-

mantic models such as Sentence-BERT (Reimers and Gurevych, 2019), represented this era. These methods can be seen as pre-trained retrievers that boosted the effectiveness of the ReQA task.

Further development led to the fusion of retrieval and generation components within the ReQA frameworks. This was implemented in systems like REALM (Guu et al., 2020) and RAG (Lewis et al., 2020b), where retrievers were co-trained with generators, further refining the performance in the ReQA task.

Recently, advanced approaches like Atlas (Izacard et al., 2022b) and RETRO (Borgeaud et al., 2022) have been introduced which could achieve performance comparable to large-scale models like Palm (Chowdhery et al., 2022) and GPT3 (Brown et al., 2020) with significantly fewer parameters.

## 3 Methodology

### 3.1 Two-Stage Training for PRCA

PRCA is designed to take sequences composed of the given query and the Top-K relevant documents retrieved by the retriever. The purpose of PRCA is to distill this collection of results, presenting a concise and effective context to the generator, while keeping both the retriever and the generator frozen. This PRCA-based paradigm introduces two challenges: the effectiveness of the retrieval cannot be directly evaluated due to its heavy dependence on the responses generated by the generator, and learning the mapping relationship between the generator's outputs and the input sequence via backpropagation is obstructed due to the black-box generator. To tackle these issues, we propose a two-stage training strategy for PRCA, as illustrated in Figure 2. In the contextual stage, supervised learning is employed to train PRCA, encouraging it to output context-rich extractions from the input text. During the reward-driven stage, the generator is treated as a reward model. The difference between the generated answer and the ground truth serves as a reward signal to further train PRCA. This process effectively optimizes the information distillation to be more beneficial for the generator to answer accurately.

### 3.2 Contextual Extraction Stage

In the contextual extraction stage, we train PRCA to extract textual information. Given an input text $S_{\text{input}}$, PRCA generates an output sequence $C_{\text{extracted}}$, representing the context derived from the

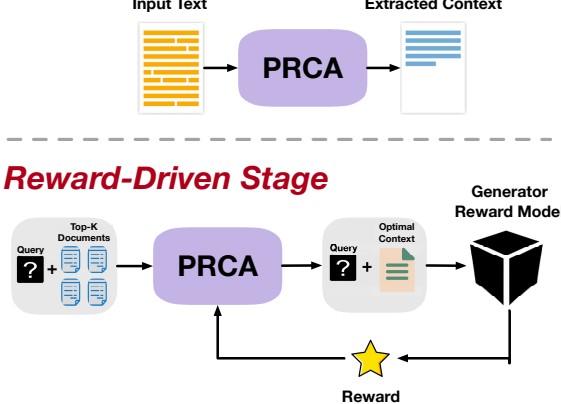

Figure 2: An illustration of the two-stage sequential training process for the PRCA. In the first "Contextual Extraction Stage", PRCA module is pre-trained on domain abstractive summarization tasks. The second "Reward-Driven Stage", demonstrates the interaction between retrieved Top-K documents and the PRCA. Here, the PRCA refines the query using both the documents and the original query, producing an optimal context. This context is processed by a generator to obtain a reward, signifying the quality and relevance of the context, with the feedback loop aiding in further refining the model's output and performance.

input text. The objective of the training process is to minimize the discrepancy between $C_{\text{extracted}}$ and the ground truth context $C_{\text{truth}}$ and the loss function is demonstrated as follows:

$$\min_{\theta} L(\theta) = -\frac{1}{N} \sum_{i=1}^{N} C_{\text{truth}}^{(i)} \log(f_{\text{PRCA}}(S_{\text{input}}^{(i)}; \theta))$$

(1)

where $\theta$ represents the parameters of PRCA

In the context extraction stage, PRCA is initialized from a BART-Large model pre-trained on CNN Daily Mail dataset (Lewis et al., 2020a).

### 3.3 Reward-Driven Stage

In the reward-driven stage, the objective is to align the extracted context $C_{\text{extracted}}$ from the previous stage with the downstream generator, ensuring that the text distilled by PRCA serves effectively to guide the generator's answering. Given the black-box nature of the generator, a direct update of PRCA is not feasible. Therefore, we resort to reinforcement learning to optimize PRCA's parameters. Specifically, the generator offers rewards to guide the update of PRCA's parameter, targeting to improve answer quality. The reward is based on the ROUGE-L score between the generated answer $O$

and the ground truth $O^*$. Meanwhile, it's vital that PRCA retains its skill of information extraction from long texts, as learned in the contextual extraction stage. Our objective is twofold: maximizing generator's reward and maintaining similarity between updated and original parameters of PRCA after contextual extraction training. Catering to the reward-driven training where policy actions manipulate sequence tokens, policy optimization, particularly via Proximal Policy Optimization (PPO) (Schulman et al., 2017; Stiennon et al., 2020), is the preferred method. However, when employing a black-box generator as a reward model, we identify certain limitations of using PPO.

In (2), we present the PPO's objective function $J(\theta)$. This function strives to optimize the advantage, a value derived from the Generalized Advantage Estimation (GAE) (Schulman et al., 2016). The GAE leverages both $\gamma$ and $\lambda$ as discounting factors, adjusting the estimated advantage based on the temporal difference $\delta_{t+l}^V$, as depicted in (3). Here, $E_t[min(r_t(\theta) \cdot A_t^{GAE}, clip(r_t(\theta), 1 - \epsilon, 1 + \epsilon) \cdot A_t^{GAE})]$ captures the expected advantage. The clip function serves to prevent excessive policy updates by constraining the policy update step, ensuring stability in the learning process. The term $\beta(V(s_t) - R_t)^2$ is a squared-error term between $V(s_t)$ and $R_t$. This term seeks to minimize the difference between the predicted and actual value, ensuring accurate value predictions. However, the critic network $V$ is usually initialized to have the same parameter as the reward model (Yao et al., 2023; Fazzie et al., 2023), which is inapplicable when the reward models are black-boxed. Additionally, the APIs from vendors usually have limited amount of return parameters which may cause the computation of $R_t$ impossible.

$$\max_\theta J(\theta) = E_t[min(r_t(\theta) \cdot A_t^{GAE},$$
$$clip(r_t(\theta), 1 - \epsilon, 1 + \epsilon) \cdot A_t^{GAE})]$$
$$- \beta(V(s_t) - R_t)^2 \qquad (2)$$

where $r_t(\theta) = \frac{\pi_\theta(a_t|s_t)}{\pi_{\theta_{ori}}(a_t|s_t)}$ is the ratio of the updated policy $\pi_\theta$ to the original policy $\pi_{\theta_{ori}}$ ; $a_t$ represents the action (the next token); $s_t$ is the state (the sequence of previous tokens); $\epsilon$ is the clipping parameter; $V$ is a critic network; $V(s_t)$ is the predicted value of state $s_t$; $\beta$ is a coefficient that weights the squared-error term; $R_t$ is the expected return at time $t$.

$$A_t^{GAE(\gamma,\lambda)} = \sum_{l=0}^{T} (\gamma\lambda)^l \delta_{t+l}^V \qquad (3)$$

where $\delta_{t+l}^V = R_{t+l} + \gamma V(s_{t+l+1}) - V(s_{t+l})$; $\gamma$ and $\lambda$ as discounting and GAE parameters respectively.

To tackle this issue, we introduce a strategy to estimate $R_t$. In the PRCA, when the token $\langle EOS \rangle$ is generated, we can obtain the reward $R_{EOS}$ by comparing the generated answer against the ground truth. We consider it an accumulation of the reward $R_t$ achieved at each time step t for the generated token. As for $R_t$, it serves as a target in $J(\theta)$ to train the critic network $V(s)$ for fitting, symbolizing the average reward of the current action, thereby assessing the advantage of the current policy. For each token, the greater the probability of generation, the more important this token is perceived by the current policy, so we consider its contribution to the total reward to be greater. Therefore, we regard the probability of generating each token as the weight of $R_{EOS}$, and the representation of $R_t$ is given by the following:

$$R_t = R_{EOS} * \frac{e^{\pi_\theta(a_t|s_t)}}{\sum_{t=1}^{K} e^{\pi_\theta(a_t|s_t)}} \qquad (4)$$

$$R_{EOS} = \text{ROUGE-L}(O, O^*)$$
$$- \beta \cdot D_{KL}(\pi_\theta || \pi_{\theta_{ori}}) \qquad (5)$$

$$\text{ROUGE-L} = \frac{\text{LCS}(X, Y)}{\max(|X|, |Y|)} \qquad (6)$$

where $K$ is the number of tokens in one generated context, $\text{LCS}(X, Y)$ denotes the length of the longest common subsequence between sequence $X$ and sequence $Y$, and $|X|$ and $|Y|$ denote the lengths of sequences $X$ and $Y$, respectively.

This method mitigates the challenges associated with calculating $R_t$ when interpreting the black-box generator as a reward model. A substantial advantage it confers is the requirement of invoking the reward model only once for each context generation. Compared to the original PPO that employs the reward model for every token computation, our approach reduces the reward model usage to $\frac{1}{K}$, which is cost-effective especially when using LLMs as generators.

Table 1: Overview of the data quantities used for training and testing across three benchmark datasets.

| Dataset | Train / Test | # of Q | # of C | # of A |
|---------|--------------|--------|--------|--------|
| SQuAD | Train | 87.6k | 18.9k | 87.6k |
|  | Test | 10.6k | 2.1k | 10.6k |
| HotpotQA | Train | 90.4k | 483.5k | 90.4k |
|  | Test | 7.4k | 66.5k | 7.4k |
| TopiQCQA | Train | 45.5k | 45.5k | 45.5k |
|  | Test | 2.5k | 2.5k | 2.5k |

Table 2: Hyperparameters settings used in the experiments.

| Hyperparameters | Value |
|-----------------|-------|
| Learning rate | $5 \times 10^{-5}$ |
| Batch size | 1/2/4 |
| Num beams | 3 |
| Temperature | 1 |
| Early Stopping | True |
| $\text{Top}_k$ | 0.0 |
| $\text{Top}_p$ | 1.0 |

## 4 Experimental Setup

### 4.1 Datasets

We performed our experiments on three QA datasets: SQuAD (Rajpurkar et al., 2016), HotpotQA (Yang et al., 2018) and TopiOCQA (Adlakha et al., 2022). The complexity of three datasets increases sequentially: SQuAD is a dataset that matches questions, documents, and answers in a one-to-one manner. HotpotQA is a multi-hop QA dataset, requiring the synthesis of correct answers from multiple documents. TopiOCQA is a conversational QA dataset with topic switching.

To align these datasets with our ReQA task, we reconstructed all three datasets into the form of $(Q, C, A)$, where $Q$ and $A$ denote the question and answer pair, and $C$ represents a corpus composed of all the documents in the dataset respectively. In Table 1, we present the number of questions and answers employed in the PRCA training and testing phases for every dataset. Additionally, we provide the quantity of documents contained within each respective corpus.

### 4.2 Baseline Retrievers and Generators

We conducted experiments with five different retrievers, specifically BM25 (Robertson and Zaragoza, 2009), SentenceBert (Reimers and Gurevych, 2019), DPR (Karpukhin et al., 2020), SimCSE (Gao et al., 2021), and Contriver (Izacard et al., 2022a). We also utilized five generators which are T5-large (Raffel et al., 2020), Phoenix-7B (Chen et al., 2023), Vicuna-7B (Peng et al., 2023), ChatGLM (Du et al., 2022) and GPT-3.5 [1] to assess the effectiveness of PRCA. Note that both the retrievers and generators remain frozen through the experiment.

By pairing every retriever with each generator,

---

[1]Our experiments were conducted with the default version of GPT-3.5-turbo and GPT-4 between May and June 2023 via https://openai.com.

we established a total of seventy-five baseline configurations on three datasets. For each configuration, we evaluated the performance with and without the application of PRCA and the difference serves as an indicator of the effectiveness of our proposed approach.

### 4.3 GPT-4 Assessment

Notably, we used GPT-4 for evaluation rather than traditional metrics like F1 and BLEU, as these metrics often misjudged semantically similar sentences. LLMs often output longer textual explanations for answers, even when the correct answer might be a word or two. Despite attempts to constrain answer lengths, the results weren't ideal. We then evaluated predictions using both manual methods and GPT-4 against golden answers. GPT-4's evaluations showed correctness rates of 96%, 93%, and 92% across three datasets, demonstrating its reliability and alignment with human judgment.

Specifically, the template for GPT-4 assessment is shown as follows. Finally, the accuracy rate of answering "Yes" is counted as the evaluation metric.

---

**Template for GPT-4 Assessment**

**Prompt**: You are now an intelligent assessment assistant. Based on the question and the golden answer, judge whether the predicted answer correctly answers the question and give only a Yes or No.
**Question:**
**Golden Answer:**
**Predicted Answer:**

---

**Expected Output:** Yes / No

---

### 4.4 Hyperparameter Configurations

To achieve optimal results in our PRCA training, careful selection of hyperparameters is pivotal. The

Table 3: Comparative results of performance for different retriever and generator combinations in the presence and absence of PRCA integration. The results are based on the evaluation using three benchmark datasets: SQuAD, HotpotQA, and TopiOCQA, and focus on the selection of the Top-5 most relevant documents.

| Retriever | Generator | SQuAD | HotpotQA | TopiOCQA |
|---|---|---|---|---|
| BM25 | T5 | 0.74**-0.03** | 0.35**+0.01** | 0.27**+0.08** |
| | Phoenix | 0.61**+0.02** | 0.31**+0.09** | 0.25**+0.03** |
| | Vicuna | 0.59**+0.09** | 0.19**+0.13** | 0.23**+0.10** |
| | ChatGLM | 0.67**+0.03** | 0.36**+0.04** | 0.35**+0.03** |
| | GPT-3.5 | 0.75**+0.02** | 0.48**+0.06** | 0.44**+0.04** |
| SentenceBert | T5 | 0.48**-0.06** | 0.20**+0.05** | 0.28**+0.05** |
| | Phoenix | 0.42**+0.04** | 0.13**+0.10** | 0.26**+0.08** |
| | Vicuna | 0.36**+0.09** | 0.22**+0.03** | 0.23**+0.05** |
| | ChatGLM | 0.57**+0.04** | 0.16**+0.08** | 0.28**+0.04** |
| | GPT-3.5 | 0.6**+0.02** | 0.34**+0.03** | 0.47**+0.03** |
| DPR | T5 | 0.57**+0** | 0.23**+0.02** | 0.20**+0.09** |
| | Phoenix | 0.56**+0.01** | 0.15**+0.09** | 0.15**+0.16** |
| | Vicuna | 0.42**+0.06** | 0.16**+0.11** | 0.15**+0.14** |
| | ChatGLM | 0.53**+0.0** | 0.16**+0.04** | 0.31**+0.07** |
| | GPT-3.5 | 0.69**+0.04** | 0.41**+0.02** | 0.34**+0.06** |
| SimSCE | T5 | 0.75**+0.01** | 0.28**+0.02** | 0.18**+0.09** |
| | Phoenix | 0.67**+0.02** | 0.17**+0.10** | 0.17**+0.13** |
| | Vicuna | 0.47**+0.06** | 0.19**+0.06** | 0.10**+0.20** |
| | ChatGLM | 0.75**+0.05** | 0.17**+0.05** | 0.21**+0.06** |
| | GPT-3.5 | 0.77**+0.04** | 0.37**+0.05** | 0.31**+0.06** |
| Contriver | T5 | 0.80**-0.08** | 0.35**+0.03** | 0.18**+0.11** |
| | Phoenix | 0.69**+0.02** | 0.10**+0.11** | 0.16**+0.18** |
| | Vicuna | 0.58**+0.08** | 0.17**+0.12** | 0.14**+0.19** |
| | ChatGLM | 0.71**+0.05** | 0.13**+0.09** | 0.23**+0.05** |
| | GPT-3.5 | 0.80**+0.02** | 0.37**+0.05** | 0.30**+0.08** |

'+' indicates an improvement in performance metrics upon the incorporation of PRCA. The color coding provides a visual representation of the effect: **Green** signifies a positive enhancement in performance, while **Red** indicates a decrement.

configuration settings employed in our experiment are stated in Table 2.

# 5 Results and Analysis

## 5.1 Overall Performance

As delineated in Table 3, among the seventy-five configurations, our experimental results suggest that the inclusion of PRCA improves performance in seventy-one configurations. On average, we observe an enhancement of 3%, 6%, and 9% on the SQuAD, HotpotQA, and TopiOCQA datasets, respectively. This demonstrates that PRCA possesses robustness and can enhance the performance of different combinations of retrievers and generators on the ReQA task. As illustrated in Figure 3, the improvements rendered by PRCA to the generators are significant across all three datasets. Particularly on the TopiOCQA dataset, the average improvement for generator Vicuna across five different retrievers reaches 14%. Notably, when SimSCE is the retriever, the enhancement offered by PRCA is 20%.

In Figure 3, we notice that the improvement to the

generator performance by PRCA across the three datasets is incremental, while the original performance of the generators across the three datasets is decremental without PRCA, correlating directly with the complexity of the datasets. This is because when faced with more complex issues, such as multi-hop questions in HotpotQA and topic transitions in multi-turn QA in TopiOCQA, PRCA reserves and integrates critical information which is beneficial for generators from the retrieved documents. This attribute of PRCA alleviate issues where generators struggle with lengthy texts, failing to answer questions correctly or producing hallucinations, thus enhancing performance.

However, the inclusion of PRCA has a negative effect on the performance of the generator T5 on the SQuAD dataset. This is because the SQuAD dataset is relatively simple, where the answer often directly corresponds to a phrase in the text. As an encoder-decoder architecture model, T5 tends to extract answers directly rather than infer in-depth based on the context. Therefore, without information distillation by PRCA from the retrieved

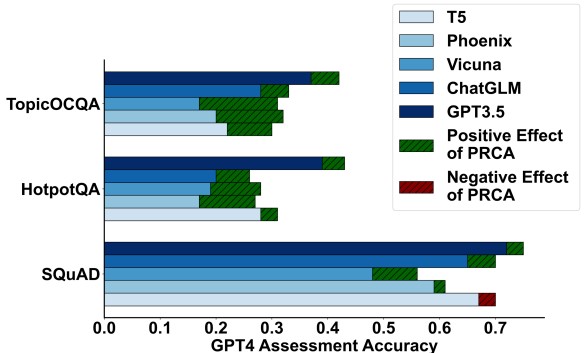

Figure 3: Comparison of performance of different generators (T5, Phoenix, Vicuna, ChatGLM, and GPT-3.5) on three benchmark datasets: SQuAD, HotpotQA, and TopicOCQA. The horizontal axis represents the GPT-4 assessment accuracy. Bars depict the performance levels of each generator, with green and red arrows indicating the enhanced or diminished effects due to PRCA integration, respectively.

documents, T5 performs well because its features fit well in handling this dataset, capable of directly extracting answers from the context. But under the effect of PRCA, the structure of the text might be altered, and T5's direct answer extraction may lead to some errors, thereby reducing performance.

While in a few configurations, the characteristics of PRCA may have negative effects, for the vast majority of configurations, our experiments validate that under PRCA-based paradigm, PRCA can effectively enhance the performance in the ReQA task, demonstrating robustness.

## 5.2 Efficiency of PRCA

PRCA represents an effective approach for enhancing the performance of the ReQA task without significantly increasing computational demand. Its efficiency is manifested in optimizing parameters to achieve superior results and in simplifying input text, thereby aiding generators in managing complex text.

**Parameter Efficiency** Figure 4 portrays a comparative analysis between the generators, which gain the maximum improvements with PRCA, and the GPT-3.5 model which operates without PRCA, across 3 datasets. PRCA boasts roughly 0.4 billion parameters, the most significantly improved generators encompass about 7 billion parameters on average, while GPT-3.5 has approximately 1.75 trillion parameters. As demonstrated in Figure 4, with a marginal

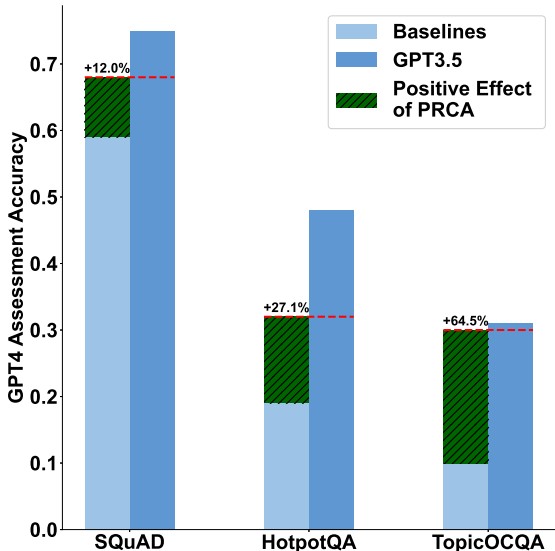

Figure 4: Performance comparison between PRCA-enhanced baseline models and GPT-3.5 across SQuAD, HotpotQA, and TopicOCQA. Light and dark blue bars represent baseline and GPT-3.5 performance, while striped green indicates PRCA's improvement.

Table 4: PRCA inference speed test results.

| Dataset | Precision | GPU | Batch Size | Inference Speed (token/s) |
|---------|-----------|------|------------|---------------------------|
| PRCA | float32 | A100 | 1 | 126 |
| PRCA | float32 | A100 | 2 | 231 |
| PRCA | float32 | A100 | 4 | 492 |

parameter increment, the performance of these generators improved by 12.0%, 27.1%, and 64.5% respectively. Hence, PRCA has great potential to be an efficient way to boost the performance of ReQA task while keeping computational resources consumption acceptable. During the inference process, a fully-trained PRCA will perform only standard forward propagation and hence introduce limited impact on inference latency. Our inference latency test on SQUAD was reported in Table 4. This low latency ensures that the system maintains a smooth process without significant delays after integrating PRCA, underscoring the high efficiency of PRCA in boosting system performance.

**Input Simplification** As illustrated in Figure 5, we analyzed the relationship between reward and token count during reward-driven stage for a QA pair in the HotpotQA dataset, with and without PRCA. There's a discernible difference in the reward trajectories with and without PRCA. Both reward curves ascend with the increase in token

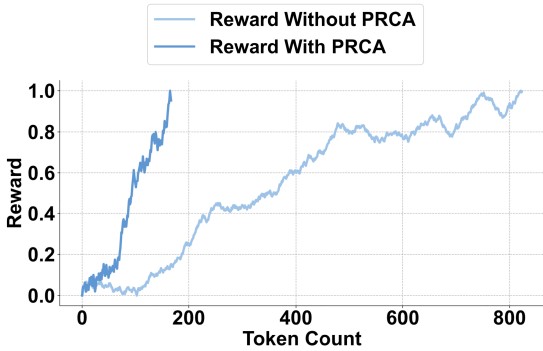

Figure 5: A depiction of reward trajectories over increasing token counts during the reward-driven stage for a QA pair within the HotpotQA dataset. Distinct lines represent rewards achieved with and without the implementation of PRCA, underscoring PRCA's ability to extract more concise and high-quality text.

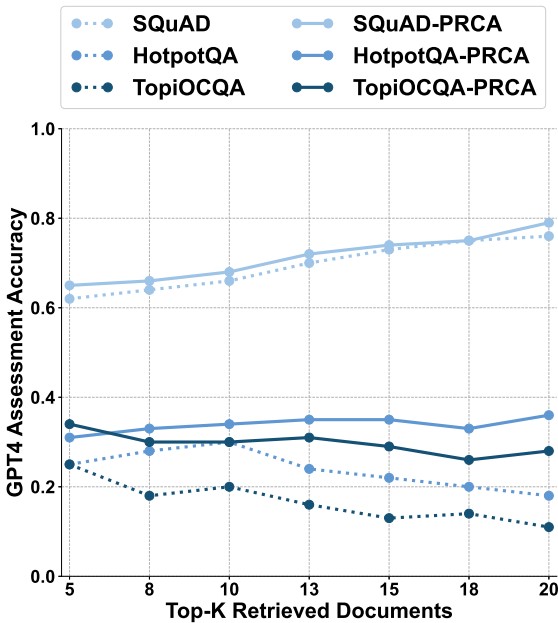

Figure 6: Comparison of performance with and without PRCA with the different number of retrieved documents.

count, but the gradient of ascent with PRCA is noticeably steeper. This implies that when PRCA is in action, the generator reaches its optimal performance with a significantly reduced token count.

Under the influence of PRCA, the generator can derive the correct answer with approximately four times fewer tokens. This indicates that PRCA can distill the retrieved text while ensuring the quality of the generated answer. This simplification process filters out redundant information, thereby promoting the generator to extract answers more accurately using a more streamlined context. Moreover, the reduction in token count enables the generator to process text faster and produce outputs more promptly. Overall, PRCA's efficiency in information distillation greatly bolsters the generator's capacity to manage and interpret complex text.

### 5.3 Impact of Top-K Selection

We conducted parameter sensitivity experiments to observe the performance of PRCA when the number of retrieved relevant documents changes. The results presented in Figure 6 show that on the SQuAD dataset, both the performance with and without PRCA improve as the number of retrieved documents increases, while the addition of PRCA consistently provides a positive effect across different Top-K values. Since the dataset is relatively simple, with the increased likelihood of the correct answer being included in the retrieved documents, both trends exhibit an upward trajectory.

In contrast, without the implementation of PRCA, there is a noticeable drop in performance on the HotpotQA and TopiOCQA datasets when more

documents are added. This decline is attributed to the model's diminishing capability to generate accurate answers to complex questions due to the rise in distracting information and the onset of hallucination problems. However, by implementing PRCA, these adverse effects are systematically alleviated, which not only reduces the onset of hallucinations but also enhances the generator's ability to handle complex queries amidst distractions.

In general, at different Top-K values, PRCA demonstrates positive effects across all three datasets, thereby illustrating the universal applicability of PRCA regardless of the quantity of retrieved documents.

### 5.4 Case Study

When answering the form of Mersenne primes problem, the retrieved text contains two distinct sources of information. One directly specifies the form as 2p-1, accurately reflecting the nature of Mersenne primes. The other source misguidedly introduces "factorial primes" as an answer. Without PRCA's intervention, this diversion leads the generator astray, resulting in an erroneous answer of "factorial primes". However, when PRCA is engaged, it sifts through the information, prioritizing the accurate context. This refined context extraction steers the generator towards the correct answer.

| Case Study without and with PRCA |
|---|
| **Question:**
Of what form do Mersenne primes take?
**Golden Answer:** 2p-1
**Part of Retrieved Documents: [Golden Answer Source]** Mersenne primes are prime numbers that are of the form 2p-1, where p is an arbitrary prime. **[Predicted Answer Source]** The Sieve of Eratosthenes, attributed to Eratosthenes, is a simple method to compute primes, although the large primes found today with computers are not generated this way. are prime. Prime numbers of this form are known as factorial primes.
**Predicted Answer without PRCA:** Factorial primes

**Context through PRCA:** Mersenne primes are prime numbers that are of the form 2p-1, where p is an arbitrary prime. The Lucas–Lehmer test is particularly fast for numbers of this form, so many of the largest primes found today are Mersenne primes.
**Predicted Answer with PRCA:** 2p-1

Note: "–" denotes key information relevant to the question, "~" represents predicted answers. |

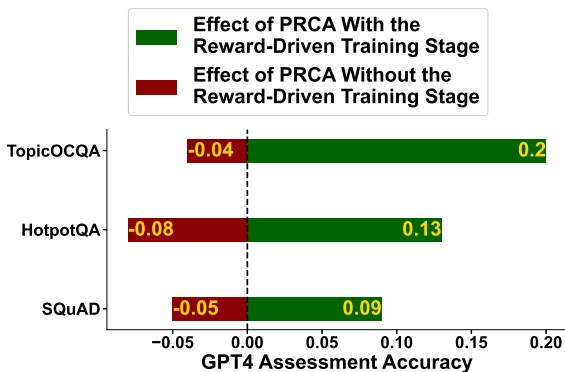

Figure 7: An illustration showcasing the impact of the reward-driven stage on PRCA's performance.

## 6 Conclusion

In conclusion, this research successfully introduces a PRCA-based paradigm for ReQA tasks, tackling the inherent challenges of fine-tuning LLMs in the retrieval-enhancement framework, especially given their vast parameter size and closed-source natures. PRCA innovatively distills retrieved documents via generator rewards, leading to a marked improvement in the ReQA task's performance. Experimental outcomes consistently demonstrate the robustness and effectiveness of PRCA when paired with various retrievers and generators, indicating its potential to be widely deployed as an adapter on the ReQA task.

## Limitations

While PRCA has shown effectiveness in improving ReQA task performance, it has limitations, including dependency on generators, convergence issues, and limited integration with retrievers. The reward during reinforcement learning training is derived from the generator, requiring PRCA retraining with different generators, which can be time-consuming. PRCA may also experience difficulties converging in a single training session, which impacts the stability and consistency of its performance. Lastly, PRCA's operation as a pluggable adapter limits its ability to train jointly with retrievers, which means if the retrieval quality is not up to par, PRCA's effectiveness could be compromised.

## Acknowledgement

Supported by the Key Research and Development Program of Guangdong Province (grant No. 2021B0101400003) and Corresponding author is Jianzong Wang (jzwang@188.com).

### 5.5 Ablation Study of PRCA

We assessed the impact of PRCA on three datasets using the configurations from section 5.2, which showed maximum improvements. The evaluation is conducted with and without the reward-driven stage to observe the impact of PRCA on the performance. As illustrated in Figure 7, without the reward-driven training stage, the effect of PRCA on the entire configuration becomes adverse because PRCA merely simplifies the text without discerning which information is beneficial for the generator to answer questions, resulting in the omission of useful text. In contrast, once the training process incorporates the reward-driven stage, the quality of the context becomes directly aligned with reward values, assisting PRCA in more effectively distilling pertinent information. Therefore, the reward-driven stage is vital, allowing PRCA to retain key details while simplifying text, enhancing its overall effect.

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
