# OpenReview forum: "PRCA: Fitting Black-Box Large Language Models for Retrieval Question Answering via Pluggable Reward-Driven Contextual Adapter"
_EMNLP/2023/Conference — EMNLP 2023 Main_

### Official Review · Reviewer_JYC9 · 2023-07-26

**Typos Grammar Style And Presentation Improvements:** Fix typos
**Soundness:** 4

**Excitement:**

3: Ambivalent: It has merits (e.g., it reports state-of-the-art results, the idea is nice), but there are key weaknesses (e.g., it describes incremental work), and it can significantly benefit from another round of revision. However, I won't object to accepting it if my co-reviewers champion it.

**Paper Topic And Main Contributions:**

This paper proposes a Pluggable Reward-driven Contextual Adapter (PRCA) to help retrieval question answering. The main contributions are:

1. PRCA allows using black-box LLMs as black boxes without fine-tuning.

2. The reinforcement learning part provides some insight to the usage of LLMs.

3. Experiments showing PRCA comprehensively improves performance on 3 QA datasets.

**Questions For The Authors:**

Q1: I didn't find any descriptions of the critic network, how is it initialized and trained?

Q2: Many experimental details are missing. How does the two-stage training work? Sequentially or iteratively? What are the hyperparameters in training such as learning rate?

Q3: The used backbones are all vanilla language models (i.e. not fine-tuned for ReQA tasks), can the proposed PRCA also work on fine-tuned baselines? Perhaps if we already have a fine-tuned generator, it can produce better results and thus less chance to be improved by external adapters like PRCA.

Q4: Line 442 mentioned that the PRCA has 4B parameters, however, Line 222 said that PRCA is initialized by pre-trained model BART-Large-CNN, which I believe is not that large. Did I miss some part of PRCA? Is it the critic network?

Q5: Following Q3, if the PRCA has 4B parameters, directly fine-tuning those while-box models which actually have a similar size to the PRCA will have a similar cost. I am curious which approach is better.

Q6: What version of GPT3.5 is used?

**Reasons To Accept:**

1. Novel method to leverage LLMs in a retrieval framework without accessing their parameters.

2. Sensible pipeline design, i.e., context extraction and reward training.

3. Providing insights to the usage of LLMs in the general QA tasks in NLP community.

**Reasons To Reject:**

1. Some details about the approach and the experiments are missing, making it hard to follow and reproduce. (See questions)

2. The authors only provide the results of some vanilla language models (e.g. Vicuna, ChatGLM, GPT-3.5) and them with PRCA.  Although I understand that the proposed method is not in competition with other specific methods, adding more baselines as references could be more informative.

3. Many improvements seem marginal in Table 2, in this case, it is suggested to conduct a significance study to confirm that the improvement brought by PRCA is indeed significant.

4. More evaluation metrics are encouraged, only using GPT4 might be biased and inaccurate.

5. As discussed in the Limitation section, the RL algorithms are usually unstable. More investigation into the stability of training can be encouraged to make the approach more applicable.

**Reproducibility:**

3: Could reproduce the results with some difficulty. The settings of parameters are underspecified or subjectively determined; the training/evaluation data are not widely available.

**Reviewer Confidence:**

4: Quite sure. I tried to check the important points carefully. It's unlikely, though conceivable, that I missed something that should affect my ratings.

---

> ### Author Rebuttal · Authors · 2023-08-29
>
> Q1: I didn't find any descriptions of the critic network, how is it initialized and trained?
>
> Response: Indeed, we have not employed a critic network in our method. Typically, the initialization of a critic network involves using the initial weights of a reward model. However, since we leverage a black-box generator as our reward model in this work, we don't have access to the critic network's initial weights.
>
> As stated from line 280 in the paper, we propose a simpler method to compute the reward for each token without requiring a critic network. Our approach innovatively estimates the reward $R_t$ for each token within the PRCA system. By harnessing the generation probability of each token, we calculate its contribution to the total reward. We obtain the primary reward, $R_{EOS}$ when an <EOS> token is produced, comparing the generated answer to the ground truth. Our method's strength lies in its efficiency: it requires the reward model's invocation just once per context generation. This is markedly more efficient than the original PPO, especially beneficial when using Large Language Models as generators.
>
> Q2: Many experimental details are missing. How does the two-stage training work? Sequentially or iteratively? What are the hyperparameters in training such as learning rate?
>
> Response: Our training process for the PRCA component is structured into two sequential stages:
> Contextual Extraction Stage: We begin by initializing the PRCA model with the weights from the BART-Large model. Following this, the model undergoes training using the question-answer pairs from our dataset.
> Reward-Driven Stage: Once the first stage is complete, we proceed to this phase where we refine the model weights through reinforcement learning, aiming to distill crucial information. It's worth noting that these stages are sequential; we complete the first before initiating the second.
> For further clarity, below is a table detailing the hyperparameters utilized during our training:
>
> | Hyperparameter  | Value               |
> |-----------------|---------------------|
> | Learning rate   | 0.00005             |
> | Batch size      | 1/2/4               |
> | num_beams       | 3                   |
> | temperature     | 1                   |
> | early_stopping  | True                |
> | top_k           | 0.0                 |
> | top_p           | 1.0                 |
>
> Q3: The used backbones are all vanilla language models (i.e. not fine-tuned for ReQA tasks), can the proposed PRCA also work on fine-tuned baselines?
>
> Response: The generator, being a black-box, can originate from diverse vendors. Consequently, there might be no feasibility or cost-effectiveness in fine-tuning them specifically for tasks. Our approach ensures adaptability even when specific task fine-tuning isn't an option.
>
> Q4: Line 442 mentioned that the PRCA has 4B parameters, however, Line 222 said that PRCA is initialized by pre-trained model BART-Large-CNN. Did I miss some part of PRCA?
>
> Response: We apologize for the oversight. The correct size of PRCA is 0.4B parameters. Line 442 contains a typing error, and we appreciate your attention to detail.
>
> Q5: Following Q3, if the PRCA has 4B parameters, directly fine-tuning those white-box models which actually have a similar size to the PRCA will have a similar cost. I am curious which approach is better.
>
> Response: Firstly, we apologize for the typing error, and I reiterate that PRCA's actual size is 0.4B. Given this, if one were to fine-tune the white-box models directly, their generally smaller parameter size would likely not reach the performance levels of the black-box models, reinforcing the relevance and utility of our approach.
>
> Q6: What version of GPT3.5 is used?
>
> Response: We utilized the gpt-3.5-turbo-0301 model, which was released by OpenAI in May 2023.
>
> Q7: As discussed in the Limitation section, the RL algorithms are usually unstable. More investigation into the stability of training can be encouraged to make the approach more applicable.
>
> Response: Trick 1 - Advantage Normalization: By normalizing the advantages, we ensure that the update step sizes are consistent, which helps in stabilizing the learning process and mitigating extreme policy updates.
>
> Trick 2 - State Normalization: This involves standardizing the state inputs to have zero mean and unit variance. By doing so, we help the policy function to generalize better and ensure smoother training trajectories.
>
> Trick 3 - Learning Rate Decay: Over the course of training, reducing the learning rate can be beneficial. As we approach the optimal policy, a smaller learning rate prevents large oscillations and helps the agent to converge to a stable solution more reliably.
>
> Thank you for your insightful questions. We believe our clarifications provide a comprehensive understanding of our approach and its advantages. We're eager to receive further feedback to refine our work.

---

### Official Review · Reviewer_N8VF · 2023-08-05

**Typos Grammar Style And Presentation Improvements:** N/A
**Soundness:** 3

**Excitement:**

3: Ambivalent: It has merits (e.g., it reports state-of-the-art results, the idea is nice), but there are key weaknesses (e.g., it describes incremental work), and it can significantly benefit from another round of revision. However, I won't object to accepting it if my co-reviewers champion it.

**Missing References:**

N/A

**Paper Topic And Main Contributions:**

This paper focused on retrieval question answering tasks by treating the Large Langauge Models (LLMs) as black box. Specifically, they propose a trainable Pluggable Reward-Driver Contextual Adapter (PRCA) between the retriever and generator. To train PRCA, there are two stages. The first stage is to extract a concise and effective context from retrieved passages with supervised loss between extracted context and ground truth context by using a BART-large model. The second stage is to use the extracted text to guide the generator's answering. They utilize ROUGE-L score as the reward to optimize the PRCA parameter while also maintain the similarity between updated and original parameters.  They conduct experiments on 3 three QA datasets with various retriever. The results show that PRCA can enhance the performance of different combinations of retrievers and generators.

**Questions For The Authors:**

See reasons to reject.

**Reasons To Accept:**

Their proposed PRCA approach achieve consistent improvement with various retriever and generation combinations. Their ablation study demonstrate the effectiveness of PCRA and reward-driven stage in PCRA.

**Reasons To Reject:**

There several questions about the papers.
1. Why do we use GPT-4 to evaluate answer accuracy? How about the common used metrics like EM and F1 in your table 2?
2. In Figure 6 and 7, which generator do you use? I am not sure the effect of PCRA if we use a strong generation like GPT-3.
3. How about the comparison between your approach the REPLUG <Replug: Retrieval-augmented 720 black-box language models>. Do you have results on their experimented datasets to have a comparison?
4. How about using ChatPGT with few-shot to as the context summarizer?

**Reproducibility:**

2: Would be hard pressed to reproduce the results. The contribution depends on data that are simply not available outside the author's institution or consortium; not enough details are provided.

**Reviewer Confidence:**

3: Pretty sure, but there's a chance I missed something. Although I have a good feel for this area in general, I did not carefully check the paper's details, e.g., the math, experimental design, or novelty.

---

> ### Author Rebuttal · Authors · 2023-08-29
>
> Thank you for your thorough review and questions. Please find our clarifications and responses below:
>
> Q1: Why do we use GPT-4 to evaluate answer accuracy? How about the common metrics like EM and F1 in your table 2?
>
> Response: EM and F1 are widely used metrics, and we initially attempted to employ them for evaluating the difference between predicted and standard answers. However, their performance was not satisfactory for our use case. A significant limitation we observed was that semantically similar sentences could have a low EM or F1 score. LLMs often output longer textual explanations for answers, even when the correct answer might be a word or two. Although we attempted to introduce prompts to constrain answer length and make EM and F1 metrics more effective, the outcomes were not as expected.
>
> Consider the example:
>
> Question: When did Tesla patent the motor?
>
> Prediction Answer: Tesla was granted the patent for the alternating current induction motor in May 1888.
>
> Golden Answer: 1888
>
> The prediction answer correctly addresses the question. Still, if we were to evaluate it using EM or F1, the score would be undeservedly low, misrepresenting the quality of the model. Hence, we adopted GPT-4-based evaluation, which excels at assessing semantic similarity between sentences.
>
> Q2: In Figure 6 and 7, which generator do you use? I am not sure about the effect of PCRA if we use a strong generation like GPT-3.
>
> Response: We employed Vicuna as the generator for Figures 6 and 7. Vicuna, as reported officially, possesses about 90% of chatGPT's capabilities, rendering it a potent generator.
>
> Furthermore, in the research paper “TopiOCQA: Open-domain Conversational Question Answering with Topic Switching” introducing the TopiQCQA dataset, experiments demonstrate that GPT-3, a strong generator, achieves only an F1 score of 31.8% on the test set. This suggests that even formidable generators might not attain optimal results on intricate datasets. In our paper's Table 2, when using GPT-3.5 as the generator, PRCA consistently showed positive effects across various datasets. We believe that PRCA is highly beneficial, even when paired with a powerful generator.
>
> Q3: How about the comparison between your approach and the REPLUG (Replug: Retrieval-augmented black-box language models)? Do you have results on their experimented datasets for comparison?
>
> Response: The dataset used by REPLUG is triviaQA. In a preliminary run of 300 contexts from triviaQA, we found the maximum text length to be 258,027 and an average length of 10,228. Given that our PRCA is trained on the BART model, there are inherent length constraints of 1024. In comparison, our three datasets offer more manageable context lengths, aligning better with our model's capacity. Meanwhile, REPLUG requires direct access to models raw outputs which might not be available from different model vendors.
>
> On the other hand, due to the closed source nature of the training data of GPT series, stronger generation baseline has a bigger potential data leakage problem and leads to less convincing experiments.
>
> Q4: How about using ChatPGT with few-shot as the context summarizer?
>
> Response: ChatGPT cannot be trained directly, so few-shot training is not feasible for it. However, if we approach this using an in-context learning method, the comparative results have already been provided in our Table 2.
>
> Thank you for your insightful questions. We believe our clarifications provide a comprehensive understanding of our approach and its advantages. We're eager to receive further feedback to refine our work.

---

### Official Review · Reviewer_bnUG · 2023-08-11

**Soundness:** 4

**Excitement:**

4: Strong: This paper deepens the understanding of some phenomenon or lowers the barriers to an existing research direction.

**Paper Topic And Main Contributions:**

The paper is about a new proposed framework  named PRCA. In this framework between the retriever and the LLM exists a generative model that improve the context for the final answer generation via LLM. The LLM and the retriever are frozen and only the "middle" model is trained

**Questions For The Authors:**

+ You state that the framework ensure efficiency , but can you provide some latency test of the pipeline
+ can you better explain why you did not use other datasets with respect to SQUAD?

**Reasons To Accept:**

+ exhaustive experiments with several baselines on different dataset
+ interesting approach to improve the LLM outcomes without finetuning the LLM itself
+ tested on different domains

**Reasons To Reject:**

+ missing further experiments on other more complex QA datasets rather than SQAD (e.g: NQ)
+ manual evaluation of the results missing
+ the trained model will learn how to optimize with respect to the final LLM
+ missing latency experiments of the pipeline with and without the context generative model

**Reproducibility:**

3: Could reproduce the results with some difficulty. The settings of parameters are underspecified or subjectively determined; the training/evaluation data are not widely available.

**Reviewer Confidence:**

3: Pretty sure, but there's a chance I missed something. Although I have a good feel for this area in general, I did not carefully check the paper's details, e.g., the math, experimental design, or novelty.

---

> ### Author Rebuttal · Authors · 2023-08-29
>
> Thank you for taking the time to review our work and provide feedback. We appreciate the detailed comments and the opportunity to address your concerns. Please find our responses to each of the issues raised below:
>
> Q1: You state that the framework ensures efficiency, but can you provide some latency test of the pipeline?
>
> Response: Regarding the efficiency of the PRCA component, we have conducted tests on the SQuAD dataset. The specific test data is as follows:
>
> | Model Data | Precision | GPU   | Batch Size | Inference Speed (token/s) |
> |---------------|------------|----------|-----------|-------------------------|
> | PRCA       | float32   | A100  | 1          | 126                       |
> | PRCA       | float32   | A100  | 2          | 231                       |
> | PRCA       | float32   | A100  | 4          | 492                       |
>
> For example, we conducted a latency test with a batch size of 1 on 100 texts. The average length was 486 tokens. On an A100 GPU, our PRCA processing speed was found to be 3.86s. This indicates that, upon integration, the PRCA component is efficient and does not introduce significant time overhead.
>
> Q2: Can you better explain why you did not use other datasets with respect to SQUAD?
>
> Response: We did consider other datasets, including NQ and triviaQA. However, there were some challenges associated with its utilization. The context lengths in NQ and triviaQA are considerably bigger. For example, in a preliminary run of 300 contexts from triviaQA, we found the maximum text length to be 258,027 and an average length of 10,228. Given that our PRCA is trained on the BART model, there are inherent length constraints of 1024. In comparison, our three datasets offer more manageable context lengths, aligning better with our model's capacity.
>
> Q3: Manual evaluation of the results is missing.
>
> Response: In fact, we have conducted manual evaluations to further validate our results. On three different datasets, we randomly selected 100 prediction-answer pairs generated by the combination of GPT-3.5 and Contriever. We then compared these predictions against the golden answers through both manual evaluation and GPT-4-based evaluation. The results indicate that GPT-4's evaluation accuracy is quite comparable to human assessment, with correctness rates of 96%, 93%, and 92% respectively. These findings suggest that GPT-4's evaluation approach can serve as a robust alternative to manual assessments, reducing the need for intensive manual work. On the other hand, GPT-4 as reported in different works, could serve as a strong evaluation benchmark.[ Vicuna: An Open-Source Chatbot Impressing GPT-4 with 90%* ChatGPT Quality,  AlpacaEval: An Automatic Evaluator of Instruction-following Models, Judging LLM-as-a-judge with MT-Bench and Chatbot Arena.]
>
> In conclusion, we believe that the clarifications and additional data provided here address the concerns raised by the reviewer. We remain committed to ensuring the robustness and validity of our approach and are open to any further feedback to improve our work.

---

### Meta-Review · Area_Chair_oT5Q · 2023-09-16

**Recommendation:** 5

**Metareview:**

**Summary:** The paper proposes to incorporate an additional module: Pluggable Reward-Driven Contextual Adapter (PRCA) in the retrieval-augmented generation setting (RAG/ReQA). PRCA is plugged between the retriever and the generator: each of the latter two modules are frozen, and only PRCA is trained (PRCA is itself based on a generative LM backbone) using a reinforcement learning approach to refine the retrieved content so as to optimize and provide a better context as prompt for the generator. Specifically, PRCA first extracts a relevant context from the retrieved passages, and then uses a ROUGE-L score to reward/penalize parameter updates while also maintaining similarity to the original parameters. The approach is empirically evaluated on three QA datasets, and beats baselines by significant margins.

All reviewers acknowledged the novelty of the solution proposed in the problem: also highlighting it's practicality to suit real world industry use-cases where the LLM cannot be trained/only API access calls are available. The claims in the paper are supported well empirically with evaluation over 3 datasets. Based on clarifications provided in the author rebuttal, most major concerns raised by the reviewers have been mitigated.

**Recommendations for improvement:** (i) One of the concerns on the paper is around the usage of only automatic evaluation (GPT-4) and no human evaluation to support the claims. The author response has partially addressed this by providing a small pilot study on 100 examples to compare human and automatic evaluation (and achieved parity in the late 90%s). I would encourage the authors to expand on this study, and add it to the main paper to strengthen claims of the paper.

(ii) The authors should add all details from the author response clarifying several of reviewer JYC9's concerns around missing details in the paper, and adding significance tests to the empirical results.

---

### Decision · Program_Chairs · 2023-10-07

**Decision:**

Accept-Main

**Comment:**

**Summary:** The paper proposes to incorporate an additional module: Pluggable Reward-Driven Contextual Adapter (PRCA) in the retrieval-augmented generation setting (RAG/ReQA). PRCA is plugged between the retriever and the generator: each of the latter two modules are frozen, and only PRCA is trained (PRCA is itself based on a generative LM backbone) using a reinforcement learning approach to refine the retrieved content so as to optimize and provide a better context as prompt for the generator. Specifically, PRCA first extracts a relevant context from the retrieved passages, and then uses a ROUGE-L score to reward/penalize parameter updates while also maintaining similarity to the original parameters. The approach is empirically evaluated on three QA datasets, and beats baselines by significant margins.

All reviewers acknowledged the novelty of the solution proposed in the problem: also highlighting it's practicality to suit real world industry use-cases where the LLM cannot be trained/only API access calls are available. The claims in the paper are supported well empirically with evaluation over 3 datasets. Based on clarifications provided in the author rebuttal, most major concerns raised by the reviewers have been mitigated.

**Recommendations for improvement:** (i) One of the concerns on the paper is around the usage of only automatic evaluation (GPT-4) and no human evaluation to support the claims. The author response has partially addressed this by providing a small pilot study on 100 examples to compare human and automatic evaluation (and achieved parity in the late 90%s). I would encourage the authors to expand on this study, and add it to the main paper to strengthen claims of the paper.

(ii) The authors should add all details from the author response clarifying several of reviewer JYC9's concerns around missing details in the paper, and adding significance tests to the empirical results.